# GBUO: "The Good, the Bad, and the Ugly" Optimizer

**Hadi Givi** [1] , **Mohammad Dehghani** [2], **Zeinab Montazeri** [2], **Ruben Morales-Menendez** [3],
**Ricardo A. Ramirez-Mendoza** [3,*] and **Nima Nouri** [4]

[1] Department of Electrical Engineering, Shahreza Campus, University of Isfahan, Iran; h.givi@shr.ui.ac.ir
[2] Department of Electrical and Electronics Engineering, Shiraz University of Technology,
 Shiraz 71557-13876, Iran; m.dehghani@sutech.ac.ir (M.D.); Z.Montazeri@sutech.ac.ir (Z.M.)
[3] School of Engineering and Sciences, Tecnologico de Monterrey, Monterrey 64849, Mexico; rmm@tec.mx
[4] Department of Electrical Engineering, Yazd University, Yazd 89195-741, Iran; nimanouri68@gmail.com
[*] Correspondence: ricardo.ramirez@tec.mx; Tel.: +52-81-2001-5597

**Abstract:** Optimization problems in various fields of science and engineering should be solved using appropriate methods. Stochastic search-based optimization algorithms are a widely used approach for solving optimization problems. In this paper, a new optimization algorithm called "the good, the bad, and the ugly" optimizer (GBUO) is introduced, based on the effect of three members of the population on the population updates. In the proposed GBUO, the algorithm population moves towards the good member and avoids the bad member. In the proposed algorithm, a new member called ugly member is also introduced, which plays an essential role in updating the population. In a challenging move, the ugly member leads the population to situations contrary to society's movement. GBUO is mathematically modeled, and its equations are presented. GBUO is implemented on a set of twenty-three standard objective functions to evaluate the proposed optimizer's performance for solving optimization problems. The mentioned standard objective functions can be classified into three groups: unimodal, multimodal with high-dimension, and multimodal with fixed dimension functions. There was a further analysis carried-out for eight well-known optimization algorithms. The simulation results show that the proposed algorithm has a good performance in solving different optimization problems models and is superior to the mentioned optimization algorithms.

**Keywords:** optimization; optimization algorithm; population-based algorithm; exploration; exploitation

## 1. Introduction

Optimization is a vital issue, which is of great importance in a wide range of applications. Generally, it can be introduced to search for the best possible solution in a feasible region of a specific problem. The main goal is to maximize the efficiency, profit, and performance of the problem. In this regard, different optimization algorithms have been applied in various fields such as energy [1,2], protection [3], energy commitment [4], electrical engineering [5–9], and energy carriers [10,11] to achieve the optimal solution.

Recently, meta-heuristic algorithms (MHAs) such as genetic algorithm (GA), particle swarm optimization (PSO), and differential evolution (DE) have been applied as powerful methods for solving various modern optimization problems. These methods have attracted researchers' attention because of their advantages such as high performance, simplicity, few parameters, avoidance of local optimization, and derivation-free mechanism. Many MHAs have been inspired by simple principles in nature, e.g., physical and biological systems. Among these algorithms, simulated annealing [12], spring search algorithm [13,14], ant colony optimization [15,16], particle swarm optimization [17], and cuckoo search [18] can be mentioned. For instance, PSO was derived based on the swarming behavior of the birds and fishes [17,19], whereas simulated annealing (SA) was proposed by considering the metal annealing process [20].

Furthermore, their appropriate mathematical models are constructed based on evolutionary concepts, intelligent biological behaviors, and physical phenomena. MHAs do not have any dependency on the nature of the problem because they utilize a stochastic approach; hence, they do not require derived information about the problem. This is counterintuitive in a mathematical method, which generally need precise information of the problem [21]. This independence from the nature of the problem is one of the main advantages of MHAs and makes them a perfect tool to find optimal solutions for an optimization problem without concern about the problem search space's nonlinearity and constraints.

Flexibility is another advantage, enabling MHAs to apply any optimization problem without changing the algorithm's main structure. These methods act as a black box with input and output modes, in which the problem and its constraints act as inputs for these methods. Hence, this characteristic makes them a potential candidate for a user-friendly optimizer.

On the other hand, contrary to mathematical methods' deterministic nature, MHAs frequently profit from random operators. As a result, compared to traditional deterministic methods, the probability of being trapped in local optimizations decreases making them independent from the initial guess.

These methods have become more prevalent since the last three decades due to their ability to quickly explore the global search space and their independence from the problem's nature. Even though, a unique benchmark does not exist to classify MHAs in the literature, the source of inspiration is one of the most popular classification criteria. Based on inspiration source, one can classify optimization algorithms into four main categories as follows: (i) swarm-based (SB), (ii) evolutionary-based (EB), (iii) physics-based (PB), and (iv) game-based (GB) algorithms. For convenience, some well-known optimization algorithms in the literature are summarized in Table 1. SB are based on simulating the behavior of living organisms, plants and natural processes, EB are based on simulation of genetic sciences, PB are designed based on simulation of various physical laws, and GB are based on simulation of different game rules [22,23].

**Table 1.** Well-known meta-heuristic algorithms (MHAs) are proposed in the literature.

| Class | Ref. | Algorithm | Main Idea (Inspiration Source) | Year |
|---|---|---|---|---|
| | [17] | Particle swarm optimization | Social behavior of birds | 1995 |
| | [24] | Cuckoo search | Behavior of cuckoo | 2009 |
| | [25] | Lion optimization algorithm | Behavior of Lion | 2016 |
| | [26] | Grasshopper optimization algorithm | Grasshopper behavior | 2017 |
| | [27] | Emperor penguin optimizer | The behavior of Emperor Penguin | 2018 |
| SB | [28] | Pity beetle algorithm | Aggregation behavior, searching for nest and food | 2018 |
| | [29] | Mouth brooding fish | The behavior of mouthbrooding Fish | 2018 |
| | [30] | Sailfish Optimizer | Group of hunting sailfish | 2019 |
| | [31] | Following Optimization Algorithm | Relationships between members and the leader of a community | 2020 |
| | [32] | Multi-Leader Optimizer | The presence of several leaders simultaneously for the population members | 2020 |
| | [33] | Genetic algorithm | Darwinian evolution theory | 1992 |
| | [34] | Differential evolution | the natural phenomenon of evolution | 1997 |
| | [35] | Genetic program | The biological model of evolution | 1998 |
| | [36] | Evolution strategy | Darwinian evolution theory | 2002 |
| EB | [37] | Biogeography-based optimizer | Biogeographic concepts | 2008 |
| | [38] | Artificial infectious disease | SEIQR epidemic model | 2016 |
| | [39] | Rooted tree optimization | Plant roots movement looking for water | 2016 |
| | [40] | Weighted superposition attraction | Weighted superposition of active fields | 2017 |

**Table 1.** *Cont.*

| Class | Ref. | Algorithm | Main Idea (Inspiration Source) | Year |
|-------|------|-----------|-------------------------------|------|
|  | [41] | Plant intelligence | Plants nervous system | 2017 |
|  | [42] | Chemotherapy science | Chemotherapy method | 2017 |
|  | [43] | Tree growth algorithm | Trees competition for acquiring light and foods | 2018 |
| PB | [44] | Simulated Annealing | Metal annealing process | 1983 |
|  | [45] | Gravitational search algorithms | Gravity law | 2009 |
|  | [46] | Water cycle algorithms | Water cycle process and how rivers and streams flow to the sea in the real world | 2012 |
|  | [47] | Water evaporation optimization | Evaporation of water molecules | 2016 |
|  | [48] | Galactic swarm optimized motion | The motion of stars, galaxies | 2016 |
|  | [49] | Spring search algorithms | Hooke's law | 2017 |
|  | [50] | Collective decision optimization | The social behavior of human beings | 2017 |
|  | [51] | Very optimistic method | Real-life practices of successful persons | 2018 |
|  | [52] | Momentum search algorithm | Momentum law and Newton's laws of motion | 2020 |
| GB | [53] | Dice game optimizer | Rules governing the game of dice and the impact of players on each other | 2019 |
|  | [54] | Orientation search algorithm | Game of orientation, in which players move in the direction of a referee | 2019 |
|  | [55] | Hide Objects Game Optimization | Behavior and movements of players to find a hidden object | 2020 |
|  | [56] | football game based optimization | Simulation of behavior of clubs in footbal league. | 2020 |
|  | [57] | Darts game optimizer | Rules of the Darts game | 2020 |
|  | [58] | Shell game optimization | Rules of the shell game | 2020 |

Each of the above-mentioned algorithms has its specific advantages and disadvantages. For instance, in thermal process which are sufficiently slow to allow time for simulation, simulated annealing guarantees that the obtained solution is optimal. Nevertheless, fine-tuning of parameters affects the convergence of the optimization problem.

In the development of MHAs, their mathematical analysis includes some open issues that require close attention. These problems are mainly of different components in MHAs that are stochastic, complex, and extremely nonlinear.

Various swarm intelligence (SI) algorithms have recently been reported. The particle swarm optimization (PSO) algorithm [17] is inspired by fishes or birds' social behavior. The artificial bee colony algorithm (ABC) [59] and the ant colony optimization (ACO) algorithm [15] are inspired by the foraging behavior of honeybees and the ants' behavior when finding the optimal path in the ant colony foraging process, respectively. The ant colony's pheromone matrix continuously evolves within the candidate solution's iteration leading to an optimal solution. This could be useful in solving path planning problems [60]. The cuckoo search algorithm (CS) [24] is a simulation of the obligate brood parasitic behavior of a certain kind of cuckoo [61]. These types of algorithms are not popular due to their high complexity. In 2011, a simulation of the cooperative foraging fruit flies' behavior was presented, resulting in the fruit fly optimization algorithm (FOA) [62]. Other examples of recently introduced SI algorithms include grey, firefly algorithm (FF) [63], wolf optimization algorithm (GWO) [64], "doctor and patient" optimization (DPO) [65], donkey theorem optimization (DTO) [66], group optimization (GO) [67], squirrel search algorithm (SSA) [68,69], dragonfly algorithm (DA) [70] among others. It is worth noting that several newly introduced MHAs, such as quasi-affine transformation evolutionary (QUATRE) [71], slime mold algorithm (SMA) [72], equilibrium optimizer (EO) [73], and Henry gas solubility (HGS) [74] show superior performance in comparison with techniques mentioned above.

QUATRE is a concurrent development framework based on quasi-affine evolution. It has been shown that this algorithm can achieve superior optimization performance for large-scale optimization problems [71,75,76]. The QUATRE algorithm can be successfully employed to extract the text feature and obtain acceptable results [32].

In recent years, the swarm intelligence algorithm as a new bionic optimization technique has been developing rapidly. However, due to the no free lunch (NFL) theorem, it is impossible to use a specific algorithm as a general method to solve all types of optimization problems [77]. The NFL theorem prompted researchers to improve classical optimization algorithms as much as possible and even introduce new algorithms to attain better performance in dealing with optimization problems.

Consequently, a novel swarm intelligence algorithm named as Harris hawks Optimization (HHO) algorithm was introduced in 2019, which is inspired by the collaborative behavior of Harris hawks in the process of hunting prey [78]. The simulation results and the performed experimentations on 29 benchmarks and different engineering optimization problems validate its high efficiency in optimization problems. The HHO algorithm has many advantages, such as few parameters adjustment, easy execution, and simple implementation. Therefore, HHO is suitable and efficient for solving practical optimization problems in many fields. For instance, it can be utilized for structure optimization [79], image segmentation [80], parameter identification [81], image denoising [82], power load distribution [83], and layout optimization [84]. It is noteworthy that, despite the attractive benefits of HHO in dealing with various optimization issues, this algorithm still has some drawbacks, namely the high complexity and the compute time consuming. In response to these problems, some scholars have proposed improvement strategies from various perspectives. For instance, introducing long-term memory into the HHO algorithm has been proposed by Hussain et al. 2019 [85], in which users are allowed to exercise based on experience, and the diversity of the population is increased.

However, disadvantages of this method include ignoring the algorithm's execution time and poor performance in high-dimensional problems. Jian et al. [80] reduced the probability of falling the HHO algorithm into a local optimum by employing dynamic control parameters and improved the global search capability by using mutation operators.

Interference terms have been added to the escape energy to control the disturbance peaks' position, enhanced by the global searchability in the next stage as reported by Fan et al. [86]. Additionally, some researchers mixed the exploration ability of other algorithms in order to improve HHO, such as simulated annealing algorithm [87], dragonfly algorithm [88], and combination of sine and cosine algorithms [89].

The main focus of the previous literature has been on the enhancement of exploratory capabilities. Meanwhile, lacking a balanced approach between search abilities leads to weakness in search results and robustness in complicated modern optimization.

In this paper, a new optimization algorithm named "the good, the bad, and the ugly" optimizer (GBUO) is proposed to solve various optimization problems. The main idea in designing GBUO is effectiveness of three population members in updating the population. GBUO is mathematically modeled and then implemented on a set of twenty-three standard objective functions.

The rest of the article is as follows: In Section 2, the proposed algorithm's steps are mathematically modeled. Simulation studies are carried out in Section 3. Then, in Section 4, the results are analyzed. Finally, in Section 5, conclusions and perspectives for future studies are presented.

## 2. "The Good, the Bad, and the Ugly" Optimizer (GBUO)

In this section, the design steps of the "the good, the bad, and the ugly" optimizer (GBUO) are explained and modeled. In GBUO, search agents scan the problem search space under the influence of three specific members of the population. Each population member is a proposed solution to the optimization problem that provides specific values

for the problem variables. Thus, the population members of an algorithm can be modeled as a matrix. The population matrix of the algorithm is specified in Equation (1).

$$X = \begin{bmatrix} X_1 \\ \vdots \\ X_i \\ \vdots \\ X_N \end{bmatrix}_{N \times m} = \begin{bmatrix} x_1^1 & \cdots & x_1^d & \cdots & x_1^m \\ \vdots & \ddots & \vdots & & \vdots \\ x_i^1 & \cdots & x_i^d & \cdots & x_i^m \\ \vdots & & \vdots & \ddots & \vdots \\ x_N^1 & \cdots & x_N^d & \cdots & x_N^m \end{bmatrix}_{N \times m}, \tag{1}$$

Here, $X$ is the population matrix, $X_i$ is the $i'$th population member, $x_i^d$ is the value for $d'$th variable specified by $i'$th member, $N$ is the number of population members, and $m$ is the number of variables.

A specific value is calculated for each population member for the objective function given that each population member represents the proposed values for the optimization problem variables. The values of the objective function are specified as a matrix in Equation (2).

$$OF = \begin{bmatrix} OF_1(X_1) \\ \vdots \\ OF_i(X_i) \\ \vdots \\ OF_N(X_N) \end{bmatrix}_{N \times 1}, \tag{2}$$

Here, $OF$ is the objective function matrix and $OF_i(X_i)$ is the value of the objective function for $i'$th population member.

The objective function's value is an indicator of whether a solution is good or bad. Based on these values, it can be determined which member provides the best quasi-optimal solution and provides the worst quasi-optimal solution. In GBUO, the algorithm's population is updated according to three members entitled good, bad, and ugly. The good is a member of the population that is the best quasi-optimal solution, and the bad is a member of the population that has presented the worst quasi-optimal solution according to the value of the objective function. Ugly is a population member that leads the algorithm's population to situations in the opposite direction. In this challenging phase, those situations of the search space that offer suitable quasi-optimal solutions are discovered. These three main members are defined in the proposed optimizer using Equations (3)–(5).

$$Good = X_g | OF_g = minimum\ of\ OF\ matrix, \tag{3}$$

$$Bad = X_b | OF_b = maximum\ of\ OF\ matrix, \tag{4}$$

$$Ugly = X_u\ and\ u\epsilon[1, 2, \ldots, N - \{g, b\}], \tag{5}$$

Here, *Good* is the good member, *Bad* is the bad member, and *Ugly* is the ugly member selected randomly.

In each algorithm iteration, the position of the population members is updated in three following phases. In the first phase, the population moves towards the good member. In the second phase, the population distances itself from the bad member. Finally, in the third phase, the ugly member leads the population to positions contrary to the population's movement. The concepts expressed in these three phases are mathematically simulated using Equations (6)–(11).

The algorithm population update is modeled based on the good member in Equations (6) and (7).

$$x_{i,nbg}^d = x_i^d + rand \times \left(Good^d - 2 \times x_i^d\right), \tag{6}$$

$$X_i = \begin{cases} X_i^{nbg}, & OF_i^{nbg} \leq OF_i \\ X_i, & else \end{cases}, \tag{7}$$

Here, $x_{i,nbg}^d$ is the new value for the $d'$th variable of $i'$th member updated based on the good member, $X_i^{nbg}$ is the new status of $i'$th member updated based on the good member, and $OF_i^{nbg}$ is the corresponding value of the objective function.

The algorithm population update is carried out based on the bad member using Equations (8) and (9).

$$x_{i,nbb}^d = x_i^d + rand \times \left( 2 \times x_i^d - Bad^d \right), \tag{8}$$

$$X_i = \begin{cases} X_i^{nbb}, & OF_i^{nbb} \leq OF_i \\ X_i, & else \end{cases}, \tag{9}$$

Here, $x_{i,nbb}^d$ is the new value for $d'$th variable of $i'$th member updated based on the bad member, $X_i^{nbb}$ is the new status of $i'$th member updated based on the bad member, and $OF_i^{nbb}$ is the corresponding value of the objective function.

The algorithm population update is modeled based on the ugly member in Equations (10) and (11).

$$x_{i,nbu}^d = x_i^d + 0.2 \times rand \times \left( Ugly^d - x_i^d \right) \times sign(OF_u - OF_i), \tag{10}$$

$$X_i = \begin{cases} X_i^{nbu}, & OF_i^{nbu} \leq OF_i \\ X_i, & else \end{cases}, \tag{11}$$

Here, $x_{i,nbu}^d$ is the new value for $d'$th variable of $i'$th member updated based on the ugly member, $sign$ denotes the sign function, $X_i^{nbu}$ represents the new status of $i'$th member updated based on the ugly member, and $OF_i^{nbu}$ is the corresponding value of the objective function.

After updating all population members based on the mentioned three phases and storing the best quasi-optimal solution, the algorithm starts the next iteration and the population members are updated by using Equations (3)–(11) and according to the new values of the objective functions. This process is repeated until the algorithm is stopped. The pseudo-code of the proposed optimizer is presented in Algorithm 1. Also, various steps of the proposed GBUO are shown as a flowchart in Figure 1.

---

**Algorithm 1. The Pseudo-Code of GBUO**

---

**Start.**
1.　　　Input information of optimization problem.
2.　　　Set parameters.
3.　　　Create an initial population.
4.　　　Calculate objective function.
5.　　　For iteration = 1:*T* *T*: maximum number of iteration
6.　　　　Update the *Good*, the *Bad*, and the *Ugly*. Equations (3)–(5).
7.　　　　For *i*=1:*N* *N*: number of population members
8.　　　　　Update $X_i$ based on the *Good*. Equations (6) and (7).
9.　　　　　Update $X_i$ based on the *Bad*. Equations (8) and (9).
10.　　　　　Update $X_i$ based on the *Ugly*. Equations (10) and (11).
11.　　　　End for *i*.
12.　　　Save the best quasi-optimal solution in this iteration.
13.　　　End for iteration.
14.　　　Output the best quasi-optimal solution of the objective function found by GBUO.
**End.**

---

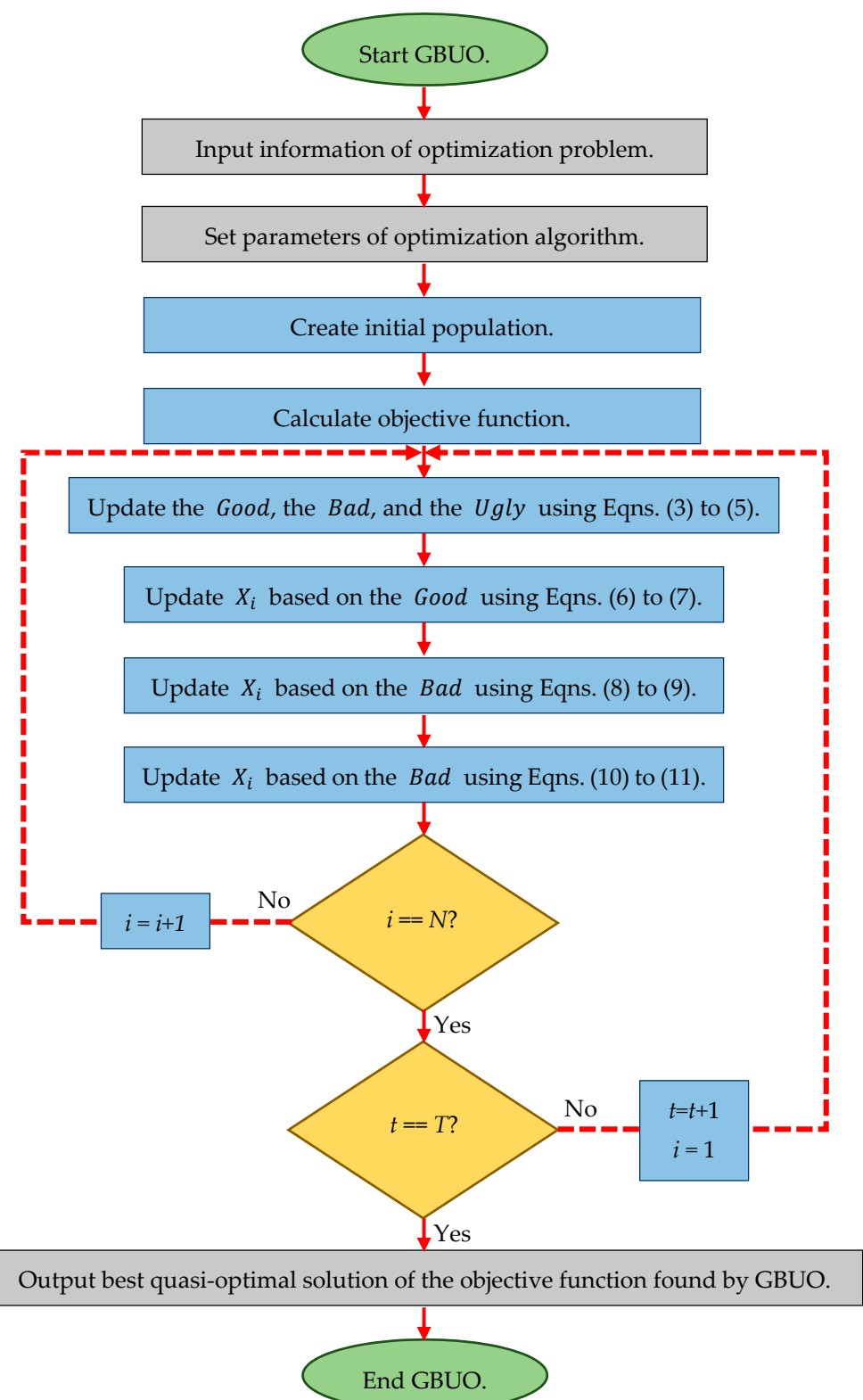

**Figure 1.** Flowchart of "the good, the bad, and the ugly" optimizer (GBUO).

## 3. Simulation Study and Results

This section evaluates GBUO performance for optimization problem resolution. For this purpose, the proposed optimizer is implemented on a set of twenty-three standard objective functions.

### 3.1. Algorithms Used for Comparison and Objective Functions

The results of other well-known optimization algorithms are compared with those obtained by GBUO in order to further evaluate its capability for solving optimization problems. These optimization algorithms are genetic algorithm (GA), particle swarm optimization (PSO), gravitational search algorithm (GSA), teaching-learning-based optimization (TLBO), grey wolf optimizer (GWO), grasshopper optimization algorithm (GOA), spotted hyena optimizer (SHO), and marine predators algorithm (MPA). The values used for the main controlling parameters of the comparative algorithms are specified in Table 2.

**Table 2.** Parameter values for the comparative algorithms.

| Algorithm | Parameter | Value |
|---|---|---|
| GA | | |
| | Type | Real coded |
| | Selection | Roulette wheel (Proportionate) |
| | Crossover | Whole arithmetic (Probability = 0.8, $\alpha = [-0.5, 1.5]$) |
| | Mutation | Gaussian (Probability = 0.05) |
| PSO | | |
| | Topology | Fully connected |
| | Cognitive and social constant | $(C_1, C_2)$ 2, 2 |
| | Inertia weight | Linear reduction from 0.9 to 0.1 |
| | Velocity limit | 10% of dimension range |
| GSA | | |
| | Alpha, $G_0$, $R_{norm}$, $R_{power}$ | 20, 100, 2, 1 |
| TLBO | | |
| | $T_F$: teaching factor | $T_F$ = round $[(1 + rand)]$ |
| | random number | *rand* is a random number between $[0 - 1]$. |
| GWO | | |
| | Convergence parameter (*a*) | *a*: Linear reduction from 2 to 0. |
| WOA | | |
| | Convergence parameter (*a*) | *a*: Linear reduction from 2 to 0. |
| | *r* is a random vector in $[0 - 1]$. | |
| | *l* is a random number in $[-1, 1]$. | |
| SHO | | |
| | Control parameter (*h*) | [5, 0]. |
| | *M* constant | [0.5, 1]. |
| MPA | | |
| | Constant number | *P*=0.5 |
| | Random vector | *R* is a vector of uniform random numbers in $[0 - 1]$. |
| | Fish Aggregating Devices (*FADs*) | *FADs*=0.2 |
| | Binary vector | *U*= 0 or 1 |

The proposed optimizer's performance and the eight optimization algorithms are evaluated for optimizing twenty-three different objective functions. These objective functions are classified into three types including unimodal, multimodal, and fixed-dimension multimodal functions. Information on these objective functions is provided in Appendix A, Tables A2, A3 and A1.

The simulation and the algorithms have been implemented in the Matlab R2020a version, run in Microsoft Windows 10 with 64 bits on a Core i-7 processor with 2.40 GHz and 6GB memory. The average (Ave) and standard deviation (std) of the best obtained optimal solution until the last iteration are computed as performance evaluation metrics. Optimization algorithms utilize 20 independent runs for each objective function, where each run employs 1000 iterations to generate and report the results.

### 3.2. Results

In this section, simulation and implementation of optimization algorithms on standard objective functions are presented. A set of seven objective functions $F_1$ to $F_7$ is introduced as

unimodal objective functions. Six objective functions $F_8$ to $F_{13}$ are considered multimodal objective functions. Finally, a set of ten objective functions $F_{14}$ to $F_{23}$ is introduced as fixed-dimension multimodal objective functions.

The optimization of the unimodal objective functions using GBUO and the mentioned eight optimization algorithms are presented in Table 3. According to the results in this table, GBUO and SHO are the best optimizers for $F_1$ to $F_4$ functions. After these two algorithms, TLBO is the third best optimizer for $F_1$ to $F_4$ functions. GBUO is also the best optimizer for $F_5$ to $F_7$ functions. Moreover, Table 4 presents the results for implementing the proposed optimizer compared with the eight optimization algorithms considered in this study for multimodal objective functions. According to this table, GBUO, SHO, MPA are the best optimizers for $F_9$ and $F_{11}$ objective functions. GBUO in $F_{10}$ function has the best performance among algorithms. After the proposed algorithm, GWO is the second and SHO is the third best optimizers for $F_{10}$. GA for $F_8$, TLBO for $F_{12}$, and GSA for $F_{13}$ are the best optimizers. GBUO is the second-best optimizer on $F_8$, $F_{12}$, and $F_{13}$. The results of applying the proposed optimizer and the eight other optimization algorithms on the third type objective functions are presented in Table 5. Based on the results in this table, GBUO provides the best performance in all $F_{14}$ to $F_{23}$ objective functions.

**Table 3.** Results of applying optimization algorithms on unimodal objective functions.

| | | GA | PSO | GSA | TLBO | GWO | GOA | SHO | MPA | GBUO |
|---|---|---|---|---|---|---|---|---|---|---|
| $F_1$ | Ave | 13.2405 | $1.7740 \times 10^{-5}$ | $2.0255 \times 10^{-17}$ | $8.3373 \times 10^{-60}$ | $1.09 \times 10^{-58}$ | $2.1741 \times 10^{-9}$ | 0 | $3.2715 \times 10^{-21}$ | 0 |
| | std | $4.7664 \times 10^{-15}$ | $6.4396 \times 10^{-21}$ | $1.1369 \times 10^{-32}$ | $4.9436 \times 10^{-76}$ | $5.1413 \times 10^{-74}$ | $7.3985 \times 10^{-25}$ | 0 | $4.6153 \times 10^{-21}$ | 0 |
| $F_2$ | Ave | 2.4794 | 0.3411 | $2.3702 \times 10^{-8}$ | $7.1704 \times 10^{-35}$ | $1.2952 \times 10^{-34}$ | 0.5462 | 0 | $1.57 \times 10^{-12}$ | 0 |
| | std | $2.2342 \times 10^{-15}$ | $7.4476 \times 10^{-17}$ | $5.1789 \times 10^{-24}$ | $6.6936 \times 10^{-50}$ | $1.9127 \times 10^{-50}$ | $1.7377 \times 10^{-16}$ | 0 | $1.42 \times 10^{-12}$ | 0 |
| $F_3$ | Ave | 1536.8963 | 589.4920 | 279.3439 | $2.7531 \times 10^{-15}$ | $7.4091 \times 10^{-15}$ | $1.7634 \times 10^{-8}$ | 0 | 0.0864 | 0 |
| | std | $6.6095 \times 10^{-13}$ | $7.1179 \times 10^{-13}$ | $1.2075 \times 10^{-13}$ | $2.6459 \times 10^{-31}$ | $5.6446 \times 10^{-30}$ | $1.0357 \times 10^{-23}$ | 0 | 0.1444 | 0 |
| $F_4$ | Ave | 2.0942 | 3.9634 | $3.2547 \times 10^{-9}$ | $9.4199 \times 10^{-15}$ | $1.2599 \times 10^{-14}$ | $2.9009 \times 10^{-5}$ | 0 | $2.6 \times 10^{-8}$ | 0 |
| | std | $2.2342 \times 10^{-15}$ | $1.9860 \times 10^{-16}$ | $2.0346 \times 10^{-24}$ | $2.1167 \times 10^{-30}$ | $1.0583 \times 10^{-29}$ | $1.2121 \times 10^{-20}$ | $28.7932$ | $9.25 \times 10^{-9}$ | 0 |
| $F_5$ | Ave | 310.4273 | 50.26245 | 36.10695 | 146.4564 | 26.8607 | 41.7767 | 28.7932 | 46.049 | 26.4322 |
| | std | $2.0972 \times 10^{-13}$ | $1.5888 \times 10^{-14}$ | $3.0982 \times 10^{-14}$ | $1.9065 \times 10^{-14}$ | 0 | $2.5421 \times 10^{-14}$ | $5.6478 \times 10^{-10}$ | 0.4219 | $3.0211 \times 10^{-15}$ |
| $F_6$ | Ave | 14.55 | 20.25 | 0 | 0.4435 | 0.6423 | $1.6085 \times 10^{-9}$ | 0.0154 | 0.398 | 0 |
| | std | $3.1776 \times 10^{-15}$ | 0 | 0 | $4.2203 \times 10^{-16}$ | $6.2063 \times 10^{-17}$ | $4.6240 \times 10^{-25}$ | 0.1784 | 0.1914 | 0 |
| $F_7$ | Ave | $5.6799 \times 10^{-3}$ | 0.1134 | 0.0206 | 0.0017 | 0.0008 | 0.0205 | $3.2915 \times 10^{-5}$ | 0.0018 | $1.5611 \times 10^{-6}$ |
| | std | $7.7579 \times 10^{-19}$ | $4.3444 \times 10^{-17}$ | $2.7152 \times 10^{-18}$ | $3.87896 \times 10^{-19}$ | $7.2730 \times 10^{-20}$ | $1.5515 \times 10^{-18}$ | $2.4384 \times 10^{-5}$ | 0.0010 | $9.0901 \times 10^{-21}$ |

**Table 4.** Results of applying optimization algorithms on multimodal objective functions.

| | | GA | PSO | GSA | TLBO | GWO | GOA | SHO | MPA | GBUO |
|---|---|---|---|---|---|---|---|---|---|---|
| $F_8$ | Ave | $-8184.4142$ | $-6908.6558$ | $-2849.0724$ | $-7408.6107$ | $-5885.1172$ | $-1663.9782$ | $-3219.7835$ | $-3594.16321$ | $-7867.6643$ |
| | Ave | 833.2165 | 625.6248 | 264.3516 | 513.5784 | 467.5138 | 716.3492 | 672.2564 | 811.32651 | 563.1864 |
| $F_9$ | Ave | 62.4114 | 57.0613 | 16.2675 | 10.2485 | $8.5265 \times 10^{-15}$ | 4.2011 | 0 | 0 | 0 |
| | std | $2.5421 \times 10^{-14}$ | $6.3552 \times 10^{-15}$ | $3.1776 \times 10^{-15}$ | $5.5608 \times 10^{-15}$ | $5.6446 \times 10^{-30}$ | $4.3692 \times 10^{-15}$ | 0 | 0 | 0 |
| $F_{10}$ | Ave | 3.2218 | 2.1546 | $3.5673 \times 10^{-9}$ | 0.2757 | $1.7053 \times 10^{-14}$ | 0.3293 | $8.8818 \times 10^{-16}$ | $9.6987 \times 10^{-12}$ | $8.8812 \times 10^{-16}$ |
| | std | $5.1636 \times 10^{-15}$ | $7.9441 \times 10^{-16}$ | $3.6992 \times 10^{-25}$ | $2.5641 \times 10^{-15}$ | $2.7517 \times 10^{-29}$ | $1.9860 \times 10^{-16}$ | $5.4216 \times 10^{-17}$ | $6.1325 \times 10^{-12}$ | $7.0652 \times 10^{-31}$ |
| $F_{11}$ | Ave | 1.2302 | 0.0462 | 3.7375 | 0.6082 | 0.0037 | 0.1189 | 0 | 0 | 0 |
| | std | $8.4406 \times 10^{-16}$ | $3.1031 \times 10^{-18}$ | $2.7804 \times 10^{-15}$ | $1.9860 \times 10^{-16}$ | $1.2606 \times 10^{-18}$ | $8.9991 \times 10^{-17}$ | 0 | 0 | 0 |
| $F_{12}$ | Ave | 0.0470 | 0.4806 | 0.0362 | 0.0203 | 0.0372 | 17414.0033 | 0.0368 | 0.0851 | 0.0328 |
| | std | $4.6547 \times 10^{-18}$ | $1.8619 \times 10^{-16}$ | $6.2063 \times 10^{-18}$ | $7.7579 \times 10^{-19}$ | $4.3444 \times 10^{-17}$ | $8.1347 \times 10^{-12}$ | $1.5461 \times 10^{-2}$ | 0.0052 | $7.1425 \times 10^{-17}$ |
| $F_{13}$ | Ave | 1.2085 | 0.5084 | 0.0020 | 0.3293 | 0.5763 | 0.3456 | 2.9575 | 0.4901 | 0.2098 |
| | std | $3.2272 \times 10^{-16}$ | $4.9650 \times 10^{-17}$ | $4.2617 \times 10^{-14}$ | $2.1101 \times 10^{-16}$ | $2.4825 \times 10^{-16}$ | $3.2539 \times 10^{-12}$ | $1.5682 \times 10^{-12}$ | 0.1932 | $1.4451 \times 10^{-16}$ |

**Table 5.** Results of applying optimization algorithms on fixed-dimension multimodal objective functions.

| | | GA | PSO | GSA | TLBO | GWO | GOA | SHO | MPA | GBUO |
|---|---|---|---|---|---|---|---|---|---|---|
| $F_{14}$ | Ave | 0.9986 | 2.1735 | 3.5913 | 2.2721 | 3.7408 | 0.9980 | 12.6705 | 0.9980 | 0.9980 |
| | std | $1.5640 \times 10^{-15}$ | $7.9441 \times 10^{-16}$ | $7.9441 \times 10^{-16}$ | $1.9860 \times 10^{-16}$ | $6.4545 \times 10^{-15}$ | $9.4336 \times 10^{-16}$ | $2.6548 \times 10^{-7}$ | $4.2735 \times 10^{-16}$ | $1.2315 \times 10^{-16}$ |
| $F_{15}$ | Ave | $5.3952 \times 10^{-2}$ | 0.0535 | 0.0024 | 0.0033 | 0.0063 | 0.0049 | 0.0003 | 0.0030 | 0.0003 |
| | std | $7.0791 \times 10^{-18}$ | $3.8789 \times 10^{-19}$ | $2.9092 \times 10^{-19}$ | $1.2218 \times 10^{-17}$ | $1.1636 \times 10^{-18}$ | $3.4910 \times 10^{-18}$ | $9.0125 \times 10^{-4}$ | $4.0951 \times 10^{-15}$ | $3.5236 \times 10^{-19}$ |
| $F_{16}$ | Ave | $-1.0316$ | $-1.0316$ | $-1.0316$ | $-1.0316$ | $-1.0316$ | $-1.0316$ | $-1.0274$ | $-1.0316$ | $-1.0316$ |
| | std | $7.9441 \times 10^{-16}$ | $3.4755 \times 10^{-16}$ | $5.9580 \times 10^{-16}$ | $1.4398 \times 10^{-15}$ | $3.9720 \times 10^{-16}$ | $9.9301 \times 10^{-16}$ | $2.6514 \times 10^{-16}$ | $4.4652 \times 10^{-16}$ | $2.4814 \times 10^{-19}$ |
| $F_{17}$ | Ave | 0.4369 | 0.7854 | 0.3978 | 0.3978 | 0.3978 | 0.4047 | 0.3991 | 0.3979 | 0.3978 |
| | std | $4.9650 \times 10^{-17}$ | $4.9650 \times 10^{-17}$ | $9.9301 \times 10^{-17}$ | $7.4476 \times 10^{-17}$ | $8.6888 \times 10^{-17}$ | $2.4825 \times 10^{-17}$ | $2.1596 \times 10^{-16}$ | $9.1235 \times 10^{-15}$ | $9.9315 \times 10^{-17}$ |
| $F_{18}$ | Ave | 4.3592 | 3 | 3 | 3.0009 | 3.0000 | 3 | 3 | 3 | 3 |
| | std | $5.9580 \times 10^{-16}$ | $3.6741 \times 10^{-15}$ | $6.9511 \times 10^{-16}$ | $1.5888 \times 10^{-15}$ | $2.0853 \times 10^{-15}$ | $5.6984 \times 10^{-15}$ | $2.6528 \times 10^{-15}$ | $1.9584 \times 10^{-15}$ | $7.7891 \times 10^{-17}$ |
| $F_{19}$ | Ave | $-3.85434$ | $-3.8627$ | $-3.8627$ | $-3.8609$ | $-3.8621$ | $-3.8627$ | $-3.8066$ | $-3.8627$ | $-3.8627$ |
| | std | $9.9301 \times 10^{-1}$ | $8.9371 \times 10^{-15}$ | $8.3413 \times 10^{-15}$ | $7.3483 \times 10^{-15}$ | $2.4825 \times 10^{-15}$ | $3.1916 \times 10^{-15}$ | $2.6357 \times 10^{-15}$ | $4.2428 \times 10^{-15}$ | $1.6512 \times 10^{-15}$ |
| $F_{20}$ | Ave | $-2.8239$ | $-3.2619$ | $-3.0396$ | $-3.2014$ | $-3.2523$ | $-3.2424$ | $-2.8362$ | $-3.3211$ | $-3.3216$ |
| | std | $3.9720 \times 10^{-16}$ | $2.9790 \times 10^{-16}$ | $2.1846 \times 10^{-14}$ | $1.7874 \times 10^{-15}$ | $2.1846 \times 10^{-15}$ | $7.9441 \times 10^{-16}$ | $5.6918 \times 10^{-15}$ | $1.1421 \times 10^{-11}$ | $1.4523 \times 10^{-17}$ |

**Table 5.** *Cont.*

|  |  | GA | PSO | GSA | TLBO | GWO | GOA | SHO | MPA | GBUO |
|---|---|---|---|---|---|---|---|---|---|---|
| $F_{21}$ | Ave | −4.3040 | −5.3891 | −5.1486 | −9.1746 | −9.6452 | −7.4016 | −4.3904 | −10.1532 | −10.1532 |
|  | std | $1.5888 \times 10^{-15}$ | $1.4895 \times 10^{-15}$ | $2.9790 \times 10^{-15}$ | $8.5399 \times 10^{-15}$ | $6.5538 \times 10^{-15}$ | $2.3819 \times 10^{-11}$ | $5.4615 \times 10^{-13}$ | $2.5361 \times 10^{-11}$ | $1.5912 \times 10^{-15}$ |
| $F_{22}$ | Ave | −5.1174 | −7.6323 | −9.0239 | −10.0389 | −10.4025 | −8.8165 | −4.6794 | −10.4029 | −10.4029 |
|  | std | $1.2909 \times 10^{-15}$ | $1.5888 \times 10^{-15}$ | $1.6484 \times 10^{-12}$ | $1.5292 \times 10^{-14}$ | $1.9860 \times 10^{-15}$ | $6.7524 \times 10^{-15}$ | $8.4637 \times 10^{-14}$ | $2.8154 \times 10^{-11}$ | $7.1512 \times 10^{-15}$ |
| $F_{23}$ | Ave | −6.5621 | −6.1648 | −8.9045 | −9.2905 | −10.1302 | −10.0003 | −3.3051 | −10.5364 | −10.5364 |
|  | std | $3.8727 \times 10^{-15}$ | $2.7804 \times 10^{-15}$ | $7.1497 \times 10^{-1}$ | $1.1916 \times 10^{-15}$ | $4.5678 \times 10^{-15}$ | $9.1357 \times 10^{-15}$ | $7.6492 \times 10^{-12}$ | $3.9861 \times 10^{-11}$ | $4.7712 \times 10^{-15}$ |

### 3.3. Statistical Testing

The optimization of standard test functions was presented as the average and standard deviation of the best solutions. However, these results alone are not enough to guarantee the superiority of the proposed algorithm. Even after twenty independent performances, this superiority may occur by chance despite its low probability. Therefore, the Friedman rank test is used to statistically evaluate the algorithms and further analyze the optimization results. The Friedman rank test is a non-parametric statistical test developed by Milton Friedman. Nonparametric means the test does not assume data comes from a particular distribution. The procedure involves ranking each row (or block) together, then considering the values of ranks by columns [90]. The steps for implementing the Friedman rank test are as follows:

Start.

Step1: Determine the results of different groups.

Step2: Rank each row of results based on the best result (here from 1 to 9).

Step3: Calculate the sum of the ranks of each column for different algorithms.

Step4: Determine the strongest algorithm to the weakest algorithm based on the sum of the ranks of each column.

End.

The Friedman rank test results for all three different objective functions: unimodal, multimodal, and fixed-dimension multimodal objective functions are presented in Table 6. Based on the results presented, for all three types of objective functions, the proposed GBUO has the first rank compared to other optimization algorithms. The overall results on all the objective functions ($F_1$–$F_{23}$) show that GBUO is significantly superior to other algorithms.

**Table 6.** Results of the Friedman rank test for evaluate the optimization algorithms.

|  | Test Function |  | GA | PSO | GSA | TLBO | GWO | GOA | SHO | MPA | GBUO |
|---|---|---|---|---|---|---|---|---|---|---|---|
| 1 | Unimodal | Friedman value | 48 | 47 | 29 | 20 | 18 | 32 | 11 | 28 | **7** |
|  | (F1–F7) | Friedman rank | 9 | 8 | 6 | 4 | 3 | 7 | 2 | 5 | **1** |
| 2 | Multimodal | Friedman value | 35 | 33 | 27 | 20 | 22 | 34 | 24 | 24 | **9** |
|  | (F8–F13) | Friedman rank | 8 | 6 | 5 | 2 | 3 | 7 | 4 | 4 | **1** |
| 3 | Fixed-dimension multimodal | Friedman value | 54 | 43 | 37 | 33 | 30 | 34 | 52 | 21 | **10** |
|  | (F14–F23) | Friedman rank | 9 | 7 | 6 | 4 | 3 | 5 | 8 | 2 | **1** |
| 4 | All 23-test functions | Friedman value | 137 | 123 | 93 | 73 | 70 | 100 | 87 | 73 | **26** |
|  |  | Friedman rank | 8 | 7 | 5 | 3 | 2 | 6 | 4 | 3 | **1** |

### 4. Discussion

Optimization algorithms based on random scanning of the search space have been widely used by researchers for solving optimization problems. Exploitation and exploration capabilities are two important indicators in the analysis of optimization algorithms. The exploitation capacity of an optimization algorithm means the ability of that algorithm to achieve and provide a quasi-optimal solution. Therefore, when comparing several optimization algorithms' performance, an algorithm that provides a more appropriate quasi-optimal solution (closer to global optimal) has a higher exploitation capacity than

other algorithms. An optimization algorithm's exploration capacity means that the algorithm's ability to accurately scan the search space, solving optimization problems with several local optimal solutions; the exploration capacity has a considerable effect on providing a quasi-optimal solution. In such problems, if the algorithm does not have the appropriate exploration capability, it provides non-optimal solutions by getting stuck in optimal local locals.

The unimodal objective functions $F_1$ to $F_7$ are functions that have only one global optimal solution and lack local optimal local. Therefore, this set of objective functions is suitable for analyzing the exploitation capacity of the optimization algorithms. Table 3 presents the results obtained from implementing the proposed GBUO and eight other optimization algorithms on the unimodal objective functions in order to properly evaluate the exploitation capacity. Evaluation of the results shows that the proposed optimizer provides more suitable quasi-optimal solutions than the other eight algorithms for all $F_1$ to $F_7$ objective functions. Accordingly, GBUO has a high exploitation capacity and is much more competitive than the other mentioned algorithms.

The second ($F_8$ to $F_{13}$) and the third ($F_{14}$ to $F_{23}$) categories of the objective functions have several local optimal solutions besides optimal solutions. Therefore, these types of objective functions are suitable for analyzing the exploration capability of the optimization algorithms. Tables 4 and 5 present the results of implementing the proposed GBUO and eight other optimization algorithms on the multimodal objective functions to tolerate capability. The results presented in these tables show that the proposed GBUO has a good exploration capability. Moreover, the proposed GBUO can also find local-optimal solutions by accurately scanning the search space and thus, does not get stuck in local optimal to the other eight algorithms. The performance of the proposed GBUO is more appropriate and competitive for solving this type of optimization problem. It is confirmed that GBUO is a useful optimizer for solving different types of optimization problems.

## 5. Conclusions

In this paper, a new optimization method called "the good, the bad, and the ugly" optimizer (GBUO) has been introduced based on the effect of three members of the population on population updating. These three influential members include the good member with the best value of the objective function, the bad member with the worst value of the objective function, and the ugly member selected randomly. In GBUO, the population is updated in three phases; in the first phase, the population moves towards the good member, in the second phase, the population moves away from the bad member, and in the third phase, the population is updated on the ugly member. In a challenging move, the ugly member leads the population to situations contrary to society's movement.

GBUO has been mathematically modeled and then implemented on a set of twenty-three different objective functions. In order to analyze the performance of the proposed optimizer in solving optimization problems, eight well-known optimization algorithms, including genetic algorithm (GA), particle swarm optimization (PSO), gravitational search algorithm (GSA), teaching-learning-based optimization (TLBO), grey wolf optimizer (GWO), whale optimization algorithm (WOA), spotted hyena optimizer (SHO), and marine predators algorithm (MPA) were considered for comparison.

The results demonstrated that the proposed optimizer has desirable and adequate performance for solving different optimization problems and is much more competitive than other mentioned algorithms.

The authors suggest some ideas and perspectives for future studies. For example, a multi-objective version of the GBUO is an exciting potential for this study. Some real-world optimization problems could be some significant contributions, as well.

**Author Contributions:** Conceptualization, M.D., Z.M. and H.G.; methodology, H.G.; software, Z.M. and M.D.; validation, R.A.R.-M. and N.N.; formal analysis, N.N., R.A.R.-M. and R.M.-M.; investigation, R.A.R.-M.; resources, Z.M. and M.D.; data curation, R.A.R.-M. and R.M.-M.; writing—original draft preparation, H.G., M.D., Z.M. and N.N.; writing—review and editing, R.A.R.-M. and R.M.-M.; supervision, M.D.; project administration, H.G.; funding acquisition, R.A.R.-M. and R.M.-M. All authors have read and agreed to the published version of the manuscript.

**Funding:** The current project was funded by Tecnologico de Monterrey and FEMSA Foundation (grant CAMPUSCITY project).

**Institutional Review Board Statement:** Not applicable.

**Informed Consent Statement:** Informed consent was obtained from all subjects involved in the study.

**Data Availability Statement:** The authors declare to honor the Principles of Transparency and Best Practice in Scholarly Publishing about Data.

**Conflicts of Interest:** The authors declare no conflict of interest.

## Appendix A

Information of the twenty-three objective functions is provided in Tables A2, A3 and A1.

**Table A1.** Unimodal test functions.

| | |
|---|---|
| $F_1(x) = \sum_{i=1}^{m} x_i^2$ | $[-100,100]^m$ |
| $F_2(x) = \sum_{i=1}^{m} |x_i| + \prod_{i=1}^{m} |x_i|$ | $[-10,10]^m$ |
| $F_3(x) = \sum_{i=1}^{m} \left( \sum_{j=1}^{i} x_i \right)^2$ | $[-100,100]^m$ |
| $F_4(x) = max\{|x_i|, \; 1 \leq i \leq m\}$ | $[-100,100]^m$ |
| $F_5(x) = \sum_{i=1}^{m-1} \left[ 100\left(x_{i+1} - x_i^2\right)^2 + (x_i - 1)^2 \right]$ | $[-30,30]^m$ |
| $F_6(x) = \sum_{i=1}^{m} ([x_i + 0.5])^2$ | $[-100,100]^m$ |
| $F_7(x) = \sum_{i=1}^{m} i x_i^4 + random(0,1)$ | $[-1.28,1.28]^m$ |

**Table A2.** Multimodal test functions

| | |
|---|---|
| $F_8(x) = \sum_{i=1}^{m} -x_i \sin(\sqrt{|x_i|})$ | $[-500,500]^m$ |
| $F_9(x) = \sum_{i=1}^{m} \left[ x_i^2 - 10\cos(2\pi x_i) + 10 \right]$ | $[-5.12,5.12]^m$ |
| $F_{10}(x) = -20\exp\left( -0.2\sqrt{\frac{1}{m}\sum_{i=1}^{m} x_i^2} \right) - \exp\left( \frac{1}{m}\sum_{i=1}^{m} cos(2\pi x_i) \right) + 20 + e$ | $[-32,32]^m$ |
| $F_{11}(x) = \frac{1}{4000}\sum_{i=1}^{m} x_i^2 - \prod_{i=1}^{m} cos\left( \frac{x_i}{\sqrt{i}} \right) + 1$ | $[-600,600]^m$ |
| $F_{12}(x) = \frac{\pi}{m}\left\{ 10\sin(\pi y_1) + \sum_{i=1}^{m}(y_i - 1)^2\left[1 + 10\sin^2(\pi y_{i+1})\right] + (y_n - 1)^2 \right\} + \sum_{i=1}^{m} u(x_i, 10, 100, 4)$ $u(x_i, a, i, n) = \begin{cases} k(x_i - a)^n & x_i > -a \\ 0 & -a < x_i < a \\ k(-x_i - a)^n & x_i < -a \end{cases}$ | $[-50,50]^m$ |
| $F_{13}(x) =$ $0.1\left\{ \sin^2(3\pi x_1) + \sum_{i=1}^{m}(x_i - 1)^2\left[1 + \sin^2(3\pi x_i + 1)\right] + (x_n - 1)^2\left[1 + \sin^2(2\pi x_m)\right] \right\} + \sum_{i=1}^{m} u(x_i, 5, 100, 4)$ | $[-50,50]^m$ |

**Table A3.** Multimodal test functions with fixed dimension

| | |
|---|---|
| $F_{14}(x) = \left( \frac{1}{500} + \sum_{j=1}^{25} \frac{1}{j + \sum_{i=1}^{2} (x_i - a_{ij})^6} \right)^{-1}$ | $[-65.53, 65.53]^2$ |
| $F_{15}(x) = \sum_{i=1}^{11} \left[ a_i - \frac{x_1 (b_i^2 + b_i x_2)}{b_i^2 + b_i x_3 + x_4} \right]^2$ | $[-5,5]^4$ |
| $F_{16}(x) = 4x_1^2 - 2.1x_1^4 + \frac{1}{3}x_1^6 + x_1 x_2 - 4x_2^2 + 4x_2^4$ | $[-5,5]^2$ |
| $F_{17}(x) = \left( x_2 - \frac{5.1}{4\pi^2} x_1^2 + \frac{5}{\pi} x_1 - 6 \right)^2 + 10 \left( 1 - \frac{1}{8\pi} \right) \cos x_1 + 10$ | $[-5,10] \times [0,15]$ |
| $F_{18}(x) = \left[ 1 + (x_1 + x_2 + 1)^2 (19 - 14x_1 + 3x_1^2 - 14x_2 + 6x_1 x_2 + 3x_2^2) \right] \times$ $\left[ 30 + (2x_1 - 3x_2)^2 \times (18 - 32x_1 + 12x_1^2 + 48x_2 - 36x_1 x_2 + 27x_2^2) \right]$ | $[-5,5]^2$ |
| $F_{19}(x) = -\sum_{i=1}^{4} c_i \exp\left( -\sum_{j=1}^{3} a_{ij} (x_j - P_{ij})^2 \right)$ | $[0,1]^3$ |
| $F_{20}(x) = -\sum_{i=1}^{4} c_i \exp\left( -\sum_{j=1}^{6} a_{ij} (x_j - P_{ij})^2 \right)$ | $[0,1]^6$ |
| $F_{21}(x) = -\sum_{i=1}^{5} \left[ (X - a_i)(X - a_i)^T + 6c_i \right]^{-1}$ | $[0,10]^4$ |
| $F_{22}(x) = -\sum_{i=1}^{7} \left[ (X - a_i)(X - a_i)^T + 6c_i \right]^{-1}$ | $[0,10]^4$ |
| $F_{23}(x) = -\sum_{i=1}^{10} \left[ (X - a_i)(X - a_i)^T + 6c_i \right]^{-1}$ | $[0,10]^4$ |

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
