# Peer review of "GBUO: “The Good, the Bad, and the Ugly” Optimizer"

_applsci, doi:10.3390/app11052042_

Round 1

Reviewer 1 Report

The paper proposes a new optimization algorithm named the Good, the Bad and the Ugly Optimizer (GBUO). It is implemented on a set of several standard objective functions to test the performance for solving optimization problems. The test of the GBUO Algorithm is demonstrated in the paper for standard objective functions classified into three groups: unimodal, multimodal with high-dimension, and multi-modal with fixed dimension functions; and results are compared with values from eight well-known optimization algorithms.

The whole paper is very well written and presented. However the style sounds sometimes very technical. While this is absolutely appropriate for expert readers, it may reduce the chance to enlarge the potential audience of the journal. That said, this is really a minor remark not affecting the overall quality of the paper.

The introduction provides adequate framework knowledge and context background to properly set the limits and objectives of the study. The adopted methodology is well structured and described within a coherent sequence of phases which are supported by tables and figures. The main methodological decisions are supported by references and are motivated according to the key objectives of the study. Conversely, the results and discussion sections could be improved.

In the Results section, would be important to describe the metrics which are being considered for the comparison between the several optimizer algorithms to demonstrate and proof the effectiveness of the proposed model. The metric is clear only for the application of the Friedman rank test. Further, some explanation and examples showing values from the tables should be included in the text.

In the Results section, would be important to describe and exemplify with values how the findings relate to the named ‘good’ (results/population move towards the good member), for the ’bad’ (results/population moves away the bad member) and ‘ugly’ (result/population moves contrary to the ugly member) equations. Further, some description and examples exemplifying how the results obtained with the GBUO algorithm and the comparison of the eight optimization algorithms would be very welcome in the results or discussion section. The metrics which are being considered for the comparison to demonstrate and proof the effectiveness of the proposed model are not clear, with exception for the application of the Friedman rank test.

Based on these comments and suggestions, the abstract could be reviewed as well. In the abstract it is mentioned that the GBUO algorithm has a performance superior to the eight well-known benchmark algorithms. Conversely, this is not exemplified with values. Perhaps, in instead of listing in the abstract the name of the eight well-known optimization algorithms employed for further analysis (GA, PSO, GSA, TLBO, GWO, WOA, SHO, MPA), the abstract could mention that ‘there was a further analysis carried-out for eight well-known optimization algorithms’ and include a description about how this test can be considered successful and show some values from the main results.

Overall, I think that the quality of the paper is very good and can be published in the journal.

Author Response

Reviewers Recommendation:

 Comments from reviewers 1:

The paper proposes a new optimization algorithm named the Good, the Bad and the Ugly Optimizer (GBUO). It is implemented on a set of several standard objective functions to test the performance for solving optimization problems. The test of the GBUO Algorithm is demonstrated in the paper for standard objective functions classified into three groups: unimodal, multimodal with high-dimension, and multi-modal with fixed dimension functions; and results are compared with values from eight well-known optimization algorithms.

The whole paper is very well written and presented. However the style sounds sometimes very technical. While this is absolutely appropriate for expert readers, it may reduce the chance to enlarge the potential audience of the journal. That said, this is really a minor remark not affecting the overall quality of the paper.

The authors appreciate dear reviewer for the carefully consideration and useful comments on the paper. It surely improves the quality of the paper. Based on these valuable comments, the article has been revised. The authors hope that the revised paper will be accepted by dear reviewer.

The introduction provides adequate framework knowledge and context background to properly set the limits and objectives of the study. The adopted methodology is well structured and described within a coherent sequence of phases which are supported by tables and figures. The main methodological decisions are supported by references and are motivated according to the key objectives of the study. Conversely, the results and discussion sections could be improved.

Response: Thank you so much to the dear reviewer for his valuable and accurate comment. The simulation results presented in the tables are discussed in the paper:

LINES: 287 to 315

“When comparing several optimization algorithms' performance, an algorithm that provides a more appropriate quasi-optimal solution (closer to global optimal) has a higher exploitation capacity than other algorithms. An optimization algorithm's exploration capacity means that the algorithm's ability to accurately scan the search space, solving optimization problems with several local optimal solutions; the exploration capacity has a considerable effect on providing a quasi-optimal solution. In such problems, if the algorithm does not have the appropriate exploration capability, it provides non-optimal solutions by getting stuck in optimal local locals.

The unimodal objective functions F1 to F7 are functions that have only one global optimal solution and lack local optimal local. Therefore, this set of objective functions is suitable for analyzing the exploitation capacity of the optimization algorithms. Table 3 presents the results obtained from implementing the proposed GBUO and eight other optimization algorithms on the unimodal objective functions in order to properly evaluate the exploitation capacity. Evaluation of the results shows that the proposed optimizer provides more suitable quasi-optimal solutions than the other eight algorithms for all F1 to F7 objective functions. Accordingly, GBUO has a high exploitation capacity and is much more competitive than the other mentioned algorithms.

The second (F8 to F13) and the third (F14 to F23) categories of the objective functions have several local optimal solutions besides optimal solutions. Therefore, these types of objective functions are suitable for analyzing the exploration capability of the optimization algorithms. Table 4 and Table 5 present the results of implementing the proposed GBUO and eight other optimization algorithms on the multimodal objective functions to tolerate capability. The results presented in these tables show that the proposed GBUO has a good exploration capability. Moreover, the proposed GBUO can also find local-optimal solutions by accurately scanning the search space and thus, does not get stuck in local optimal to the other eight algorithms. The performance of the proposed GBUO is more appropriate and competitive for solving this type of optimization problem. It is confirmed that GBUO is a useful optimizer for solving different types of optimization problems.”

And LINES 278 to 283

“The Friedman rank test results for all three different objective functions: unimodal, multimodal, and fixed-dimension multimodal objective functions are presented in Table 6. Based on the results presented, for all three types of objective functions, the proposed GBUO has the first rank compared to other optimization algorithms. The overall results on all the objective functions (F1-F23) show that GBUO is significantly superior to other algorithms.”

In the Results section, would be important to describe the metrics which are being considered for the comparison between the several optimizer algorithms to demonstrate and proof the effectiveness of the proposed model. The metric is clear only for the application of the Friedman rank test. Further, some explanation and examples showing values from the tables should be included in the text.

Response: Thank you so much to the dear reviewer for his valuable and accurate comment. This test has been used in some similar articles recently published by other researchers and authors.

For this reason, the authors have used the Friedman rank test in this article.

However, the optimization results presented in Tables 1 to 4 indicate the superiority of the proposed algorithm. This test is mostly presented to prove that the results are non-random.

LINES: 244 to 245

“The average (Ave) and standard deviation (std) of the best obtained optimal solution until the last iteration are computed as performance evaluation metrics.”

In the Results section, would be important to describe and exemplify with values how the findings relate to the named ‘good’ (results/population move towards the good member), for the ’bad’ (results/population moves away the bad member) and ‘ugly’ (result/population moves contrary to the ugly member) equations. Further, some description and examples exemplifying how the results obtained with the GBUO algorithm and the comparison of the eight optimization algorithms would be very welcome in the results or discussion section. The metrics which are being considered for the comparison to demonstrate and proof the effectiveness of the proposed model are not clear, with exception for the application of the Friedman rank test.

Response: Thank you so much to the dear reviewer for his valuable and accurate comment. This comment significantly improves the quality of the article. To address this valuable comment parameter values for the comparative algorithms have been added.

LINES: 235 to 236 and 248 to 249

The values used for the main controlling parameters of the comparative algorithms are specified in Table 2.

Table 2. Parameter values for the comparative algorithms.

Algorithm

parameter

value

GA

Type

Real coded

Selection

Roulette wheel (Proportionate)

Crossover

Whole arithmetic (Probability = 0.8,
)

Mutation

Gaussian (Probability = 0.05)

PSO

Topology

Fully connected

Cognitive and social constant

(C1, C2) 2, 2

Inertia weight

Linear reduction from 0.9 to 0.1

Velocity limit

10% of dimension range

GSA

Alpha, G0, Rnorm, Rpower

20, 100, 2, 1

TLBO

TF: teaching factor

TF = round  

random number

rand is a random number between

GWO

Convergence parameter (a)

a: Linear reduction from 2 to 0.

WOA

Convergence parameter (a)

a: Linear reduction from 2 to 0.

r is a random vector in

l is a random number in

SHO

Control parameter (h)

M constant

MPA

Constant number

P=0.5

Random vector

R is a vector of uniform random numbers in

Fish Aggregating Devices (FADs)

????=0.2

Binary vector

U= 0 or 1

Based on these comments and suggestions, the abstract could be reviewed as well. In the abstract it is mentioned that the GBUO algorithm has a performance superior to the eight well-known benchmark algorithms. Conversely, this is not exemplified with values. Perhaps, in instead of listing in the abstract the name of the eight well-known optimization algorithms employed for further analysis (GA, PSO, GSA, TLBO, GWO, WOA, SHO, MPA), the abstract could mention that ‘there was a further analysis carried-out for eight well-known optimization algorithms’ and include a description about how this test can be considered successful and show some values from the main results.

Response: Thank you so much to the dear reviewer for his valuable and accurate comment. Based on this valuable comment this sentence of abstract has been improved.

LINES: 24 to 33

“The mentioned standard objective functions can be classified into three groups: unimodal, multimodal with high-dimension, and multimodal with fixed dimension functions. There was a further analysis carried-out for eight well-known optimization algorithms. The simulation results show that the proposed algorithm has a good performance in solving different optimization problems models and is superior to the mentioned optimization algorithms.”

Overall, I think that the quality of the paper is very good and can be published in the journal.

The authors appreciate dear reviewer for the carefully consideration and useful comments on the paper. It surely improves the quality of the paper. Based on these valuable comments, the article has been revised. The authors hope that the revised paper will be accepted by dear reviewer.

Reviewer 2 Report

I think this paper is an interesting article, and analytical results sounds with rational implications. However, there are some minor comments need to be corrected before being accepted.

1. This study compares a variety of optimization algorithms, including genetic algorithm (GA), particle swarm optimization (PSO), gravitational search algorithm (GSA), optimization based on teaching and research (TLBO), gray wolf optimizer (GWO), whale optimization algorithm ( WOA), Spotted Hyena Optimizer (SHO) and Marine Predator Algorithm (MPA), the research process is rigorous, and the conclusions are quite reasonable.

2. However, although the Friedman rank test is used for comparison in Table 5, it is not statistically significant, and it is impossible to further compare the significance of the difference.

3. Can the meaning of the variables in the matrix be explained further? And how does this information use these 8 optimization algorithms?

4. Authors should detail further on how the optimization algorithms was parameterized.

5. Can the author perform calculations and comparisons through real data?

6. The conclusion could be more specific and direct. I would like to see some explicit information on the literature gaps, possible contributions in introduction.

Author Response

Reviewers Recommendation:

Comments from reviewers 2:

I think this paper is an interesting article, and analytical results sounds with rational implications. However, there are some minor comments need to be corrected before being accepted.

  1. This study compares a variety of optimization algorithms, including genetic algorithm (GA), particle swarm optimization (PSO), gravitational search algorithm (GSA), optimization based on teaching and research (TLBO), gray wolf optimizer (GWO), whale optimization algorithm (WOA), Spotted Hyena Optimizer (SHO) and Marine Predator Algorithm (MPA), the research process is rigorous, and the conclusions are quite reasonable.

The authors appreciate dear reviewer for the carefully consideration and useful comments on the paper. It surely improves the quality of the paper. Based on these valuable comments, the article has been revised. The authors hope that the revised paper will be accepted by dear reviewer.

  1. However, although the Friedman rank test is used for comparison in Table 5, it is not statistically significant, and it is impossible to further compare the significance of the difference.

Response: Thank you so much to the dear reviewer for his valuable and accurate comment. This test has been used in some similar articles recently published by other researchers and authors.

For this reason, the authors have used the Friedman rank test in this article.

However, the optimization results presented in Tables 1 to 4 indicate the superiority of the proposed algorithm. This test is mostly presented to prove that the results are non-random.

The authors hope that these results will be accepted by the dear reviewer for the intended purpose (ensure that the superiority of the algorithm is not random).

  1. Can the meaning of the variables in the matrix be explained further? And how does this information use these 8 optimization algorithms?

Response: Thank you so much to the dear reviewer for his valuable and accurate comment. Based on this valuable comment the meaning of the variables in the matrix have been explained.

Each population-based optimization algorithm has a population that is represented using a matrix called the population matrix. Each row of the population matrix represents a population member that provides values for the problem variables. Based on the values proposed by each member of the population, the objective function of the problem is evaluated.

This population matrix and objective function values are the same in all eight other algorithms.

LINES: 165 to 179

“In GBUO, search agents scan the problem search space under the influence of three specific members of the population.Each population member is a proposed solution to the optimization problem that provides specific values for the problem variables. Thus, the population members of an algorithm can be modeled as a matrix. The population matrix of the algorithm is specified in Eqn. (1).

,

(1)

Here,  is the population matrix,  is the i’th population member,  is the value for d'th variable specified by i'th member,  is the number of population members, and  is the number of variables.

A specific value is calculated for each population member for the objective function given that each population member represents the proposed values for the optimization problem variables. The values of the objective function are specified as a matrix in Eqn. (2).

,

(2)

Here,  is the objective function matrix and  is the value of the objective function for i’th population member.”

  1. Authors should detail further on how the optimization algorithms was parameterized.

Response: Thank you so much to the dear reviewer for his valuable and accurate comment. This comment significantly improves the quality of the article. To address this valuable comment parameter values for the comparative algorithms have been added.

LINES: 235 to 236 and 248 to 249

The values used for the main controlling parameters of the comparative algorithms are specified in Table 2.

Table 2. Parameter values for the comparative algorithms.

Algorithm

parameter

value

GA

Type

Real coded

Selection

Roulette wheel (Proportionate)

Crossover

Whole arithmetic (Probability = 0.8,
)

Mutation

Gaussian (Probability = 0.05)

PSO

Topology

Fully connected

Cognitive and social constant

(C1, C2) 2, 2

Inertia weight

Linear reduction from 0.9 to 0.1

Velocity limit

10% of dimension range

GSA

Alpha, G0, Rnorm, Rpower

20, 100, 2, 1

TLBO

TF: teaching factor

TF = round  

random number

rand is a random number between

GWO

Convergence parameter (a)

a: Linear reduction from 2 to 0.

WOA

Convergence parameter (a)

a: Linear reduction from 2 to 0.

r is a random vector in

l is a random number in

SHO

Control parameter (h)

M constant

MPA

Constant number

P=0.5

Random vector

R is a vector of uniform random numbers in

Fish Aggregating Devices (FADs)

????=0.2

Binary vector

U= 0 or 1

  1. Can the author perform calculations and comparisons through real data?

Response: Thank you so much to the dear reviewer for his valuable and accurate comment. As presented in the conclusion section, the implementation of the proposed algorithm can be applied to various real optimization problems.

“The authors suggest some ideas and perspectives for future studies. For example, a multi-objective version of the GBUO is an exciting potential for this study. Some re-al-world optimization problems could be some significant contributions, as well.”

One of the special potentials of the proposed GBUO algorithm is its use in solving various real optimization problems in different sciences. To this end, the authors will make the source code of the GBUO available to readers and researchers at the same time as the possible publication of this article.

  1. The conclusion could be more specific and direct. I would like to see some explicit information on the literature gaps, possible contributions in introduction.

Response: Thank you so much to the dear reviewer for his valuable and accurate comment.

In the conclusion section, the authors have tried to present a comprehensive conclusion and also in the last paragraph of the article, to point out the possible future works and studies of this study.

However, based on this valuable comment, the authors have removed parts of the conclusion section.

The authors ask the dear reviewer that the conclusion section be unchanged, but if the dear reviewer does not agree with the authors, the authors will act on the opinion of the dear reviewer.

LINES: 317 to 349

“5. Conclusions

In this paper, a new optimization method called “the Good, the Bad, and the Ugly” Optimizer (GBUO) has been introduced based on the effect of three members of the population on population updating. These three influential members include the good member with the best value of the objective function, the bad member with the worst value of the objective function, and the ugly member selected randomly. In GBUO, the population is updated in three phases; in the first phase, the population moves towards the good member, in the second phase, the population moves away from the bad member, and in the third phase, the population is updated on the ugly member. In a challenging move, the ugly member leads the population to situations contrary to society's movement.

GBUO has been mathematically modeled and then implemented on a set of twenty-three different objective functions. In order to analyze the performance of the proposed optimizer in solving optimization problems, eight well-known optimization algorithms, including Genetic Algorithm (GA), Particle Swarm Optimization (PSO), Gravitational Search Algorithm (GSA), Teaching-Learning-Based Optimization (TLBO), Grey Wolf Optimizer (GWO), Whale Optimization Algorithm (WOA), Spotted Hyena Optimizer (SHO), and Marine Predators Algorithm (MPA) were considered for comparison.

The results demonstrated that the proposed optimizer has desirable and adequate performance for solving different optimization problems and is much more competitive than other mentioned algorithms.

The authors suggest some ideas and perspectives for future studies. For example, a multi-objective version of the GBUO is an exciting potential for this study. Some real-world optimization problems could be some significant contributions, as well.”

The authors have also added several changes to the introduction.

LINES: 80 to 83

“SB are based on simulating the behavior of living organisms, plants and natural processes, EB are based on simulation of genetic sciences, PB are designed based on simulation of various physical laws, and GB are based on simulation of different game rules [22, 23].”

LINES: 154 to 158

“In this paper, a new optimization algorithm named “the Good, the Bad, and the Ugly” Optimizer (GBUO) is proposed to solve various optimization problems. The main idea in designing GBUO is effectiveness of three population members in updating the population. GBUO is mathematically modeled and then implemented on a set of twenty-three standard objective functions.”

Reviewer 3 Report

In this study, the new optimization algorithm was proposed. Authors clearly characterized their own optimization algorithm and showed that in comparison to other well known algorithms, proposed solutions have better performance. 

The study was prepared quite well, the proposed solution was described in detail. Very interesting is the section with comparison of the GBUO performence with other  algorithms, but this section could be more detailed, plase see my comments:

  • keywords: the keywords should contain the words not mentioned in the title of paper, please improve this,
  • line 60: ad- 59 it is "advantages of Mas", should be "advantages of MAs"
  • Please clearly state the aim of the work at the end of the introduction.
  • Table 2-5, Heading of last column - it is GPUO it should be GBUO.
  • Section 3.3. - please give more detail for Friedman rank test, describe how it works, 
  • The results presented in table 2-5 should be wider discussed, 

Author Response

Reviewers Recommendation:

Comments from reviewers 3:

In this study, the new optimization algorithm was proposed. Authors clearly characterized their own optimization algorithm and showed that in comparison to other well known algorithms, proposed solutions have better performance. 

The study was prepared quite well, the proposed solution was described in detail. Very interesting is the section with comparison of the GBUO performence with other  algorithms, but this section could be more detailed, plase see my comments:

The authors appreciate dear reviewer for the carefully consideration and useful comments on the paper. It surely improves the quality of the paper. Based on these valuable comments, the article has been revised. The authors hope that the revised paper will be accepted by dear reviewer.

  1. keywords: the keywords should contain the words not mentioned in the title of paper, please improve this,

Response: Thank you so much to the dear reviewer for his valuable and accurate comment. Based on this valuable comment, the authors have been improved the keywords.

“Keywords: optimization; optimization algorithm; population-based algorithm; exploration; exploitation”

  1. line 60: ad- 59 it is "advantages of Mas", should be "advantages of MAs"

Response: Thank you so much to the dear reviewer for his valuable and accurate comment. This issue has been resolved.

(NOTE: in revised version of manuscript: “MHAs” has been used instead ”MAs”. )

LINES: 59

“This independence from the nature of the problem is one of the main advantages of MHAs and …”

  1. Please clearly state the aim of the work at the end of the introduction.

Response: Thank you so much to the dear reviewer for his valuable and accurate comment. Based on this valuable comment, the authors have added article's purpose before the last sentence.

LINES: 154 to 158

“In this paper, a new optimization algorithm named “the Good, the Bad, and the Ugly” Optimizer (GBUO) is proposed to solve various optimization problems. The main idea in designing GBUO is effectiveness of three population members in updating the population. GBUO is mathematically modeled and then implemented on a set of twenty-three standard objective functions.”

  1. Table 2-5, Heading of last column - it is GPUO it should be GBUO.

Response: Thank you so much to the dear reviewer for his valuable and accurate comment. This issue has been resolved. Based on this valuable comment GPUO has been changed to GBUO in these tables.

  1. Section 3.3. - please give more detail for Friedman rank test, describe how it works, 

Response: Thank you so much to the dear reviewer for his valuable and accurate comment. This test has been used in some similar articles recently published by other researchers and authors.

For this reason, the authors have used the Friedman rank test in this article.

However, the optimization results presented in Tables 1 to 4 indicate the superiority of the proposed algorithm. This test is mostly presented to prove that the results are non-random.

The authors hope that these results will be accepted by the dear reviewer for the intended purpose (ensure that the superiority of the algorithm is not random).

  1. The results presented in table 2-5 should be wider discussed, 

Response: Thank you so much to the dear reviewer for his valuable and accurate comment. The simulation results presented in the tables are discussed in the paper:

LINES: 287 to 315

“When comparing several optimization algorithms' performance, an algorithm that provides a more appropriate quasi-optimal solution (closer to global optimal) has a higher exploitation capacity than other algorithms. An optimization algorithm's exploration capacity means that the algorithm's ability to accurately scan the search space, solving optimization problems with several local optimal solutions; the exploration capacity has a considerable effect on providing a quasi-optimal solution. In such problems, if the algorithm does not have the appropriate exploration capability, it provides non-optimal solutions by getting stuck in optimal local locals.

The unimodal objective functions F1 to F7 are functions that have only one global optimal solution and lack local optimal local. Therefore, this set of objective functions is suitable for analyzing the exploitation capacity of the optimization algorithms. Table 3 presents the results obtained from implementing the proposed GBUO and eight other optimization algorithms on the unimodal objective functions in order to properly evaluate the exploitation capacity. Evaluation of the results shows that the proposed optimizer provides more suitable quasi-optimal solutions than the other eight algorithms for all F1 to F7 objective functions. Accordingly, GBUO has a high exploitation capacity and is much more competitive than the other mentioned algorithms.

The second (F8 to F13) and the third (F14 to F23) categories of the objective functions have several local optimal solutions besides optimal solutions. Therefore, these types of objective functions are suitable for analyzing the exploration capability of the optimization algorithms. Table 4 and Table 5 present the results of implementing the proposed GBUO and eight other optimization algorithms on the multimodal objective functions to tolerate capability. The results presented in these tables show that the proposed GBUO has a good exploration capability. Moreover, the proposed GBUO can also find local-optimal solutions by accurately scanning the search space and thus, does not get stuck in local optimal to the other eight algorithms. The performance of the proposed GBUO is more appropriate and competitive for solving this type of optimization problem. It is confirmed that GBUO is a useful optimizer for solving different types of optimization problems.”

And LINES 278 to 283

“The Friedman rank test results for all three different objective functions: unimodal, multimodal, and fixed-dimension multimodal objective functions are presented in Table 6. Based on the results presented, for all three types of objective functions, the proposed GBUO has the first rank compared to other optimization algorithms. The overall results on all the objective functions (F1-F23) show that GBUO is significantly superior to other algorithms.”

Reviewer 4 Report

Interesting article in the subject of Applied Sciences journal

The article presents a new solution of one meta-heuristics algorithms procedure

The article needs some review to improve its readability

I have some advice for authors presented in my comments.

Comment 1

At the end of the Introduction you wrote:

“The main focus of the previous literature has been on the enhancement of exploratory capabilities. Meanwhile, lacking a balanced approach between search abilities leads to weakness in search results and robustness in complicated modern optimization.

The rest of the article is as follows: In Section 2, the proposed algorithm's steps are mathematically modeled. Simulation studies are carried out in Section 3. Then, in Section 4, the results are analyzed. Finally, in Section 5, conclusions and perspectives for future studies are presented.”

I propose adding the article's purpose before the last sentence. For example, in Abstract you present this sentence

“In this paper, a new optimization algorithm called “the 16Good, the Bad, and the Ugly” Optimizer (GBUO) is introduced, based on the effect of three members of the population on the population updates.”

Comment 2

You wrote chapter 2

2.“. The Good, the Bad, and the Ugly” Optimizer (GBUO)

Please improve it on:

  1. “The Good, the Bad, and the Ugly” Optimizer (GBUO)

Comment 3

In chapter 2 you started from

“In this section, the design steps of the proposed optimizer are explained and modeled. In GBUO, search agents scan the problem search space under the influence of three specific members of the population. Each population member is a proposed solution to the optimization problem that provides specific values for the problem variables. Thus, the population members of an algorithm can be modeled as a matrix. The population matrix of the algorithm is specified in Eqn. (1).”

You use short GBUO without its description only in the title of chapter reader can know what is meaning of this short.

I propose to add an explanation of GBUO short

Comment 4

In an Introduction, you present many procedures and propose a short for it. Among others

Meta-heuristics algorithms (MAs). You use this short in many places in the article and also you many times write full name Meta-heuristics algorithms.

If author agree I propose to use another short

Meta-heuristics algorithms (MHAs). In my opinion, it better shows the meaning of word meta-heuristics

I found in Internet:

"Heuristics find 'good' solutions on large-size problem instances. They allow to obtain acceptable performance at acceptable costs in a wide range of problems. They do not have an approximation guarantee on the obtained solutions. They are tailored and designed to solve a specific problem or/and instance.

Meta-heuristics are general-purpose algorithms that can be applied to solve almost any optimization problem. They may be viewed as upper-level general methodologies that can be used as a guiding strategy in designing underlying heuristics...”

Comment 5

Line 59

This independence from the nature of the problem is one of the main advantages of Mas and makes them a perfect tool to find optimal solutions for an optimization problem without concern about the problem search space's nonlinearity and constraints.

Please improve Mas on MAs  or if you change with my advice on MHAs

This independence from the nature of the problem is one of the main advantages of MAs and makes them a perfect tool to find optimal solutions for an optimization problem without concern about the problem search space's nonlinearity and constraints.

Comment 6

In many places, you use meta-heuristic algorithms after you propose short MAs

Line 48

Many meta-heuristic algorithms have been inspired by simple principles in nature, e.g., physical and biological systems.

Many MAs have been inspired by simple principles in nature,e.g., physical and biological systems.

In line 87

In the development of meta-heuristics algorithms, their mathematical analysis includes some open issues that require close attention. These problems are mainly of different components in meta-heuristic algorithms that are stochastic, complex, and extremely nonlinear.

Comment 7

Table 4. Results of applying optimization algorithms on fixed-dimension multimodal objective functions.

Table 4 is not readable in std parameter

I propose a decrease of font size used for write this table

Comment 8

Table 5. Results of the Friedman rank test for evaluate the optimization algorithms

To improve the readability of this table I propose to enlarge second column

Author Response

Reviewers Recommendation:

Comments from reviewers 4:

Interesting article in the subject of Applied Sciences journal

The article presents a new solution of one meta-heuristics algorithms procedure

The article needs some review to improve its readability

I have some advice for authors presented in my comments.

The authors appreciate dear reviewer for the carefully consideration and useful comments on the paper. It surely improves the quality of the paper. Based on these valuable comments, the article has been revised. The authors hope that the revised paper will be accepted by dear reviewer.

  1. Comment 1

At the end of the Introduction you wrote:

“The main focus of the previous literature has been on the enhancement of exploratory capabilities. Meanwhile, lacking a balanced approach between search abilities leads to weakness in search results and robustness in complicated modern optimization.

The rest of the article is as follows: In Section 2, the proposed algorithm's steps are mathematically modeled. Simulation studies are carried out in Section 3. Then, in Section 4, the results are analyzed. Finally, in Section 5, conclusions and perspectives for future studies are presented.”

I propose adding the article's purpose before the last sentence. For example, in Abstract you present this sentence

“In this paper, a new optimization algorithm called “the 16Good, the Bad, and the Ugly” Optimizer (GBUO) is introduced, based on the effect of three members of the population on the population updates.”

Response: Thank you so much to the dear reviewer for his valuable and accurate comment. Based on this valuable comment, the authors have added article's purpose before the last sentence.

LINES: 154 to 158

“In this paper, a new optimization algorithm named “the Good, the Bad, and the Ugly” Optimizer (GBUO) is proposed to solve various optimization problems. The main idea in designing GBUO is effectiveness of three population members in updating the population. GBUO is mathematically modeled and then implemented on a set of twenty-three standard objective functions.”

  1. Comment 2

You wrote chapter 2

2.“. The Good, the Bad, and the Ugly” Optimizer (GBUO)

Please improve it on:

  1. “The Good, the Bad, and the Ugly” Optimizer (GBUO)

Response: Thank you so much to the dear reviewer for his valuable and accurate comment. This issue has been resolved.

LINE: 163

2.“The Good, the Bad, and the Ugly” Optimizer (GBUO)

  1. Comment 3

In chapter 2 you started from

“In this section, the design steps of the proposed optimizer are explained and modeled. In GBUO, search agents scan the problem search space under the influence of three specific members of the population. Each population member is a proposed solution to the optimization problem that provides specific values for the problem variables. Thus, the population members of an algorithm can be modeled as a matrix. The population matrix of the algorithm is specified in Eqn. (1).”

You use short GBUO without its description only in the title of chapter reader can know what is meaning of this short.

I propose to add an explanation of GBUO short

“the Good, the Bad, and the Ugly” Optimizer (GBUO)

Response: Thank you so much to the dear reviewer for his valuable and accurate comment. To address this valuable comment, the authors have been added explanation of GBUO.

LINES: 164 to 165

“In this section, the design steps of the “the Good, the Bad, and the Ugly” Optimizer (GBUO) are explained and modeled.”

  1. Comment 4

In an Introduction, you present many procedures and propose a short for it. Among others

Meta-heuristics algorithms (MAs). You use this short in many places in the article and also you many times write full name Meta-heuristics algorithms.

If author agree I propose to use another short

Meta-heuristics algorithms (MHAs). In my opinion, it better shows the meaning of word meta-heuristics

I found in Internet:

"Heuristics find 'good' solutions on large-size problem instances. They allow to obtain acceptable performance at acceptable costs in a wide range of problems. They do not have an approximation guarantee on the obtained solutions. They are tailored and designed to solve a specific problem or/and instance.

Meta-heuristics are general-purpose algorithms that can be applied to solve almost any optimization problem. They may be viewed as upper-level general methodologies that can be used as a guiding strategy in designing underlying heuristics...”

Response: Thank you so much to the dear reviewer for his valuable and accurate comment. Based on this valuable suggestion, the authors have been used “MHAs” instead ”MA”.  Also, in other cases where the full name of "Meta-heuristics algorithms" was used, the abbreviation of that word "MHAs" was replaced based on the suggestion of the dear reviewer.

These changes have been highlighted in the text of the manuscript.

  1. Comment 5

Line 59

This independence from the nature of the problem is one of the main advantages of Mas and makes them a perfect tool to find optimal solutions for an optimization problem without concern about the problem search space's nonlinearity and constraints.

Please improve Mas on MAs  or if you change with my advice on MHAs

This independence from the nature of the problem is one of the main advantages of MAs and makes them a perfect tool to find optimal solutions for an optimization problem without concern about the problem search space's nonlinearity and constraints.

Response: Thank you so much to the dear reviewer for his valuable and accurate comment. Based on this valuable suggestion, the authors have been used “MHAs” instead ”MA”.  Also, in other cases where the full name of "Meta-heuristics algorithms" was used, the abbreviation of that word "MHAs" was replaced based on the suggestion of the dear reviewer.

Recently, meta-heuristic algorithms (MHAs) such as Genetic Algorithm (GA), Particle Swarm Optimization (PSO), and Differential Evolution (DE) have been applied as powerful methods for solving various modern optimization problems. These methods have attracted researchers' attention because of their advantages such as high performance, simplicity, few parameters, avoidance of local optimization, and derivation-free mechanism. Many MHAs have been inspired …

Furthermore, their appropriate mathematical models are constructed based on evolutionary concepts, intelligent biological behaviors, and physical phenomena. MHAs do not have any dependency on the nature of the problem because they utilize a stochastic approach; hence, they do not require derived information about the problem. This is counterintuitive in a mathematical method, which generally need precise information of the problem. This independence from the nature of the problem is one of the main advantages of MHAs and makes them a perfect tool to find optimal solutions for an optimization problem without concern about the problem search space's nonlinearity and constraints.

Flexibility is another advantage, enabling MHAs to apply any optimization problem without changing the algorithm's main structure. These methods act as a black box with input and output modes, in which the problem and its constraints act as inputs for these methods. Hence, this characteristic makes them a potential candidate for a user-friendly optimizer.

On the other hand, contrary to mathematical methods' deterministic nature, MHAs frequently profit from random operators.

Even though, a unique benchmark does not exist to classify MHAs in the literature, the source of inspiration is one of the most popular classification criteria.

Table 1. Well-known MHAs are proposed in the literature.

In the development of MHAs, their mathematical analysis includes some open issues that require close attention. These problems are mainly of different components in MHAs that are stochastic, complex, and extremely nonlinear.

It is worth noting that several newly introduced MHAs, such as quasi …

  1. Comment 6

In many places, you use meta-heuristic algorithms after you propose short MAs

Line 48

Many meta-heuristic algorithms have been inspired by simple principles in nature, e.g., physical and biological systems.

Many MAs have been inspired by simple principles in nature,e.g., physical and biological systems.

In line 87

In the development of meta-heuristics algorithms, their mathematical analysis includes some open issues that require close attention. These problems are mainly of different components in meta-heuristic algorithms that are stochastic, complex, and extremely nonlinear.

Response: Thank you so much to the dear reviewer for his valuable and accurate comment. Based on this valuable suggestion, the authors have been used “MHAs” instead ”MAs”.  Also, in other cases where the full name of "Meta-heuristics algorithms" was used, the abbreviation of that word "MHAs" was replaced based on the suggestion of the dear reviewer.

Many MHAs have been inspired by simple principles in nature,e.g., physical and biological systems.

And

In the development of MHAs, their mathematical analysis includes some open issues that require close attention. These problems are mainly of different components in MHAs that are stochastic, complex, and extremely nonlinear.

  1. Comment 7

Table 4. Results of applying optimization algorithms on fixed-dimension multimodal objective functions.

Table 4 is not readable in std parameter

I propose a decrease of font size used for write this table

Response: Thank you so much to the dear reviewer for his valuable and accurate comment. In order to better display the results, all tables, especially Table 4, have been organized.

  1. Comment 8

Table 5. Results of the Friedman rank test for evaluate the optimization algorithms

To improve the readability of this table I propose to enlarge second column

Response: Thank you so much to the dear reviewer for his valuable and accurate comment. In order to better display the results, all tables, especially Table 4 and Table 5, have been organized.

(NOTE: in revised version: Table 4 changed to Table 5 and Table 5 changed to Table 6)

Table 4. Results of applying optimization algorithms on fixed-dimension multimodal objective functions.

GPUO

MPA

SHO

GOA

GWO

TLBO

GSA

PSO

GA

0.9980

0.9980

12.6705

0.9980

3.7408

2.2721

3.5913

2.1735

0.9986

Ave

F14

1.2315E-16

4.2735E-16

2.6548E-07

9.4336E-16

6.4545E-15

1.9860E-16

7.9441E-16

7.9441E-16

1.5640E-15

std

0.0003

0.0030

0.0003

0.0049

0.0063

0.0033

0.0024

0.0535

5.3952E-02

Ave

F15

3.5236E-19

4.0951E-15

9.0125E−04

3.4910E-18

1.1636E-18

1.2218E-17

2.9092E-19

3.8789E-19

7.0791E-18

std

-1.0316

-1.0316

-1.0274

-1.0316

-1.0316

-1.0316

-1.0316

-1.0316

-1.0316

Ave

F16

2.4814E-16

4.4652E-16

2.6514E-16

9.9301E-16

3.9720E-16

1.4398E-15

5.9580E-16

3.4755E-16

7.9441E-16

std

0.3978

0.3979

0.3991

0.4047

0.3978

0.3978

0.3978

0.7854

0.4369

Ave

F17

9.9315E-17

9.1235E-15

2.1596E-16

2.4825E-17

8.6888E-17

7.4476E-17

9.9301E-17

4.9650E-17

4.9650E-17

std

3

3

3

3

3.0000

3.0009

3

3

4.3592

Ave

F18

7.7891E-17

1.9584E-15

2.6528E-15

5.6984E-15

2.0853E-15

1.5888E-15

6.9511E-16

3.6741E-15

5.9580E-16

std

-3.8627

-3.8627

-3.8066

-3.8627

-3.8621

-3.8609

-3.8627

-3.8627

-3.85434

Ave

F19

1.6512E-15

4.2428E-15

2.6357E-15

3.1916E-15

2.4825E-15

7.3483E-15

8.3413E-15

8.9371E-15

9.9301E-17

std

-3.3216

-3.3211

-2.8362

-3.2424

-3.2523

-3.2014

-3.0396

-3.2619

-2.8239

Ave

F20

1.4523E-17

1.1421E-11

5.6918E-15

7.9441E-16

2.1846E-15

1.7874E-15

2.1846E-14

2.9790E-16

3.97205E-16

std

-10.1532

-10.1532

-4.3904

-7.4016

-9.6452

-9.1746

-5.1486

-5.3891

-4.3040

Ave

F21

1.5912E-15

2.5361E-11

5.4615E-13

2.3819E-11

6.5538E-15

8.5399E-15

2.9790E-16

1.4895E-15

1.5888E-15

std

-10.4029

-10.4029

-4.6794

-8.8165

-10.4025

-10.0389

-9.0239

-7.6323

-5.1174

Ave

F22

7.1512E-15

2.8154E-11

8.4637E-14

6.7524E-15

1.9860E-15

1.5292E-14

1.6484E-12

1.5888E-15

1.2909E-15

std

-10.5364

-10.5364

-3.3051

-10.0003

-10.1302

-9.2905

-8.9045

-6.1648

-6.5621

Ave

F23

4.7712E-15

3.9861E-11

7.6492E-12

9.1357E-15

4.5678E-15

1.1916E-15

7.1497E-14

2.7804E-15

3.8727E-15

std

Table 5. Results of the Friedman rank test for evaluate the optimization algorithms.

GPUO

MPA

SHO

GOA

GWO

TLBO

GSA

PSO

GA

Test function

7

28

11

32

18

20

29

47

48

Friedman value

Unimodal

(F1-F7)

1

1

5

2

7

3

4

6

8

9

Friedman rank

9

24

24

34

22

20

27

33

35

Friedman value

Multimodal

(F8-F13)

2

1

4

4

7

3

2

5

6

8

Friedman rank

10

21

52

34

30

33

37

43

54

Friedman value

Fixed-dimension multimodal

(F14-F23)

3

1

2

8

5

3

4

6

7

9

Friedman rank

26

73

87

100

70

73

93

123

137

Friedman value

All 23-test functions

4

1

3

4

6

2

3

5

7

8

Friedman rank

Round 2

Reviewer 3 Report

Dear Authors, 

Thank you very much for your effort and provided changes. 

Almost all my remarks was adressed, excpet this about Friedman rank test and wider discussion of the results. So I am claryfying my previous comments.

  • please insert some short description of Friedman rank test, I know that it was described elswhere, but for clarity and better understanding of the content one or two sentences bout friedman rank test should be added in the text,
  • the discussion of the results should be extended with the with existing literature.

Best regards

Author Response

Reviewers Recommendation:

Comments from reviewers 3:

Dear Authors, 

Thank you very much for your effort and provided changes. 

Almost all my remarks was adressed, excpet this about Friedman rank test and wider discussion of the results. So I am claryfying my previous comments.

The authors appreciate dear reviewer for the carefully consideration and useful comments on the paper. It surely improves the quality of the paper. Based on these valuable comments, the article has been revised. The authors hope that the revised paper will be accepted by dear reviewer.

  • please insert some short description of Friedman rank test, I know that it was described elswhere, but for clarity and better understanding of the content one or two sentences bout friedman rank test should be added in the text,

Response: Thank you so much to the dear reviewer for his valuable and accurate comment. This comment significantly improves the quality of the article. Base on this valuable comment, explanatory sentences as well as steps of implementing the Friedman rank test have been added. 

LINES: 279 to 290

“The Friedman rank test is a non-parametric statistical test developed by Milton Friedman. Nonparametric means the test doesn’t assume data comes from a particular distribution. The procedure involves ranking each row (or block) together, then considering the values of ranks by columns [90]. The steps for implementing the Friedman rank test are as follows:

Start.

Step1: Determine the results of different groups.

Step2: Rank each row of results based on the best result (here from 1 to 9).

Step3: Calculate the sum of the ranks of each column for different algorithms.

Step4: Determine the strongest algorithm to the weakest algorithm based on the sum of the ranks of each column.

End.”

  • the discussion of the results should be extended with the with existing literature.

Response: Thank you so much to the dear reviewer for his valuable and accurate comment. Based on this valuable comment, the discussion of the results has been extended.

LINES: 250 to 263

“The optimization of the unimodal objective functions using GBUO and the mentioned eight optimization algorithms are presented in Table 3. According to the results in this table, GBUO and SHO are the best optimizers for F1 to F4 functions. After these two algorithms, TLBO is the third best optimizer for F1 to F4 functions. GBUO is also the best optimizer for F5 to F7 functions. Moreover, Table 4 presents the results for implementing the proposed optimizer compared with the eight optimization algorithms considered in this study for multimodal objective functions. According to this table, GBUO, SHO, MPA are the best optimizers for F9 and F11 objective functions. GBUO in F10 function has the best performance among algorithms. After the proposed algorithm, GWO is the second and SHO is the third best optimizers for F10. GA for F8, TLBO for F12, and GSA for F13 are the best optimizers. GBUO is the second-best optimizer on F8, F12, and F13. The results of applying the proposed optimizer and the eight other optimization algorithms on the third type objective functions are presented in Table 5. Based on the results in this table, GBUO provides the best performance in all F14 to F23 objective functions.”

And LINES: 301 to 305

“Optimization algorithms based on random scanning of the search space have been widely used by researchers for solving optimization problems. Exploitation and exploration capabilities are two important indicators in the analysis of optimization algorithms. The exploitation capacity of an optimization algorithm means the ability of that algorithm to achieve and provide a quasi-optimal solution.”
